# ACTIVE PROBABILISTIC DRUG DISCOVERY

## ABSTRACT

Early drug discovery plays a crucial role in the development of new medications by focusing on the identification and optimization of lead molecules that specifically bind to target proteins. However, this process is accompanied by various challenges, such as the vastness of molecule libraries, high attrition rate, and the intricate nature of molecular interactions. To overcome these challenges, there is a paradigm shift towards integrating intelligence and automation into end-to-end operations. Intelligent computing aids in the discovery and recommendation of molecules, while automated experiments offer data validation and feedback. This innovative approach can be viewed as an active probabilistic learning problem, assuming that active molecules binding to a specific target are typically a small proportion and exhibit cluster-distributed characteristics. Based on this formulation, we propose a novel active probabilistic drug discovery (APDD) method, which iteratively updates the binding probabilities of molecules to progressively enhance drug discovery performance with three consecutive steps of probabilistic clustering, selective docking, and active wet-experiment. We conduct extensive experiments on two benchmark datasets of DUD-E and LIT-PCBA and a simulated virtual library. The results demonstrate the feasibility and efficiency of our approach, showcasing substantial cost savings with an average reduction of 80% in computational docking expenses and 70% in wet experimental costs, while maintaining high accuracy in lead molecule discovery.

## 1 INTRODUCTION

Early drug discovery plays a critical role in the pharmaceutical industry and scientific community (Hughes et al., 2011). It encompasses the application of computational drug design and chemical biology techniques to effectively identify and optimize lead molecules. Computational drug design utilizes knowledge of the target's structure or known active ligands to aid in the identification of potential candidate drugs, employing either a structure-based or ligand-based approach depending on available information (Yu & MacKerell, 2017). On the other hand, chemical biology focuses on comprehending the mechanism of action of a chemical modulator on a specific target (Grigalunas et al., 2020). Nonetheless, the intricate behavior of drug molecules poses substantial challenges in early drug discovery, leading to high failure rates and significant hurdles to be overcome (Ngwewondo et al., 2021).

In recent years, there has been a strong focus on integrating intelligent computing and automated experimentation into accelerating the drug discovery process (Schneider, 2018; Wei et al., 2023; Ivanenkov et al., 2023). The objective is to explore a wider range of chemical possibilities while reducing costs. Intelligent computing plays a crucial role by recommending molecules for further investigation (Wei et al., 2023). By analyzing vast amounts of data, such as chemical structures and biological assays, intelligent computing can identify potential drug candidates (Wei et al., 2023) that are more likely to succeed. This not only speeds up the process of identifying promising molecules but also reduces the need for extensive laboratory experiments (Wei et al., 2023). Automated experimentation is another important component of this integrated approach (Schneider, 2018). Through high-throughput screening techniques and the use of automated technologies, researchers can efficiently test numerous molecules against specific targets or disease models (Schneider, 2018). The results from automated experiments validate the predictions made by intelligent computing and help refine the models (Schneider, 2018). The continuous feedback loop between computational predictions and experimental results enables researchers to learn and optimize their hypotheses. This

iterative cycle allows for more efficient and targeted drug discovery, reducing the time and resources required to develop innovative therapies (Wei et al., 2023).

To achieve this end-to-end early drug discovery, the design of a novel machine learning framework plays an essential role (Yoo et al., 2023). From the aspect of machine learning, this new paradigm of early drug discovery can be formulated as an active probabilistic learning problem. Specifically, it is based on the assumption that the number of active molecules capable of binding to a target protein of interest within a molecule library is typically relatively small and clustered within the chemical space (Singh et al., 2023). probabilistic clustering algorithms can be employed to identify these desirable clusters, and representative molecules can be selected from these clusters through active learning techniques. Automated experiments are conducted on the selected molecules, and the feedback generated by these experiments is used to optimize the drug discovery models (Wei et al., 2023). This iterative process of experiment and feedback helps to refine the model, leading to more accurate predictions and a higher likelihood of discovering effective drug candidates.

Based on this formulation, we propose a novel approach termed Active Probabilistic Drug Discovery (APDD), which comprises three primary components. Firstly, molecules are partitioned into clusters using probabilistic clustering algorithms with substructure features. This clustering method helps identify relevant chemical features and group molecules based on their similarities. Second, representative molecules from each cluster are selected for molecular docking simulations using Vina-GPU+ (Ding et al., 2023). These simulations aim to identify clusters featuring molecules with high docking probabilities, which serve as potential candidates for further experimentation. Finally, a few molecules from these clusters are selected through active learning techniques to undergo next wet experimental screening to provide feedback for the model. The feedback generated by these experiments is integrated into the model to refine its predictions and improve its performance iteratively.

To evaluate the effectiveness of our proposed APDD method, comprehensive experiments were conducted on 90 targets from well-established benchmark datasets DUD-E (Mysinger et al., 2012) and LIT-PCBA (Tran-Nguyen et al., 2020), which are widely used for early drug discovery evaluations. The experimental results showcased that the APDD method demonstrated substantial computational cost savings by an impressive 80% on average, compared to traditional drug discovery methods. Furthermore, the APDD method achieved remarkable reductions in wet experimental costs, with an average reduction of 70%. Despite these significant cost reductions, the APDD method maintained comparable screening accuracy[1]. We also evaluate APDD for virtual screening with simulation experiments over 1.4 million molecules, which further validate the feasibility and efficiency of APDD. To sum up, the primary contributions of this work lie in formulating this new paradigm for early drug discovery and designing a novel machine learning algorithm that is effective and can easily be extended to accommodate an extra-large virtual molecule library.

## 2 RELATED WORK

Automated systems have played a significant role in early drug discovery over the years (Schneider, 2018). Medium-to-high throughput drug screening in specialized assays has become commonplace in the pharmaceutical industry, resulting in increased laboratory efficiency, lower overall attrition rates, and reduced costs throughout the drug discovery value chain (Shinn et al., 2019). On the other side, the introduction of intelligent computing-driven early drug discovery has further accelerated the growth of the pharmaceutical sector, leading to a revolutionary change in the industry (Pasrija et al., 2022). Moving forward, an ideal approach for early drug discovery would involve the integration of intelligent computing and automated systems (Schneider, 2018; Wei et al., 2023; Ivanenkov et al., 2023). This integration has huge potential to expedite drug development and approval process, ultimately enabling therapies to reach patients more quickly (Wei et al., 2023). The benefits of this approach are numerous (Schneider, 2018). By applying standardized procedures with automated support, measurement errors can be minimized, and material consumption can be reduced. Additionally, the use of automated systems allows for shorter screening and testing cycles, enabling rapid feedback loops and molecule optimization. Moreover, this approach enables molecular optimization based on multiple relevant biochemical and biological endpoints without personal bias (Schneider, 2018). In short, considering the vast size of the chemical space, estimated to be between 1030 and

---

[1]The code and data will be freely available upon acceptance.

1060 drug-like molecules, a key challenge for medicinal chemists is determining "what to make and test next" (Schneider, 2018). Automated and intelligent drug discovery platforms must be capable of providing accurate answers to this question (Schneider, 2018).

Currently, numerous research institutions and pharmaceutical companies are actively engaged in research and development in this field (Wei et al., 2023; Ivanenkov et al., 2023). In academia, for example, Moreira-Filho et al. (2021) highlighted the recent advancements in automating whole organism screening, target-based assays, fragment-based drug discovery (FBDD), computer-aided drug design (CADD), and the integration of intelligent computing tools in drug design projects, marking a new era in the discovery of anti-schistosomiasis drugs (Moreira-Filho et al., 2021). Grisoni et al. (2021) proposed a design-make-test-analyze framework as a blueprint for automated drug design, utilizing intelligent computing and miniaturized bench-top synthesis, and successfully applied this framework to explore the chemical space of known LXR$\alpha$ agonists and generate novel molecular candidates (Grisoni et al., 2021). In the industrial sector, for instance, XtalPi Ltd. developed the ID4Inno drug discovery platform (Wei et al., 2023). This innovative "trinity" paradigm combines artificial intelligence, automated experiments, and expert knowledge to explore a broader chemical space with increased efficiency and reduced costs (Wei et al., 2023). The platform has been validated by the discovery of potent hit molecules against PI5P4K-$\beta$, a novel anti-cancer target (Wei et al., 2023). Insilico Medicine Inc. launched Pharma. AI, a fully automated AI-driven drug discovery platform (Ivanenkov et al., 2023). This platform demonstrates efficient identification of novel molecular structures targeting DDR1 and CDK20 (Ivanenkov et al., 2023). Collaborating with Strateos, Inc., Eli Lilly has introduced a web-based platform that provides broad access to the world's first cloud laboratory[2]. It integrates automated biology and chemistry research capabilities, creating a closed-loop system that accelerates the design-make-test-analyze drug discovery cycle.

## 3 FORMULATION

Early drug discovery involves various initial steps to identify molecules that exhibit desirable characteristics for developing effective drugs. Early drug discovery will typically rely on a combination of intelligence and automated systems to facilitate seamless operations and enhance the overall performance of the methods employed. Artificial intelligence plays a crucial role in the discovery and recommendation of potential molecules, while automated experiments ensure data validation and provide valuable feedback. From a machine learning perspective, this innovative paradigm to early drug discovery can be formulated as an active probabilistic learning problem. Firstly, this paradigm is built on the assumption that within a compound library, the number of active molecules capable of binding to a specific target protein of interest is generally limited and tends to be concentrated within a specific region of the chemical space (Singh et al., 2023). Secondly, the outcomes of early drug discovery typically yield a list of diverse molecules with the highest affinity. However, the hit rate of the top-ranked molecules remains limited, ranging from only 10-35% (Gorgulla et al., 2020; Sadybekov et al., 2022; Gorgulla et al., 2023). Studies on the progression from hits to clinical drug candidates have shown that hit molecules, on average, share only 50% structural similarity with the final drugs (Brown, 2023). Also, it is widely recognized that computational docking affinity scores struggle to achieve a strong correlation with experimental results due to the complex nature of the physical processes involved. Consequently, it becomes more crucial for prioritizing a list of molecules based on the highest binding probabilities than solely considering their affinity scores. Thirdly, the utilization of probability clustering algorithms allows for the identification of these desirable clusters, from which representative molecules can be selected using active learning techniques. Finally, the selected molecules undergo automated experiments, and the feedback obtained from these experiments is utilized to optimize the drug discovery models.

## 4 METHOD

In this section, we will present a comprehensive overview of the construction process for the Active Probabilistic Drug Discovery (APDD) framework. We will commence by elucidating the methodology employed for probabilistic clustering of a ligand molecule library, leveraging substructure features as outlined in Section 4.1. Subsequently, in Section 4.2, we will delve into the intricacies

---

[2]https://strateos.com/medicinal-chemistry/.

associated with the utilization of Vina-GPU+ for the docking process of representative molecules from each cluster. Moving forward, Section 4.3 will introduce our query strategy, which actively selects molecules, alongside an explanation of how we refine docking probabilities by incorporating wet lab results, ultimately amplifying the effectiveness of our drug discovery efforts.

## 4.1 MOLECULE CLUSTERING

Molecule Clustering plays an important role in reducing the computation cost of traditional Drug Discovery methods, as traversing through millions of molecules remains prohibitively time-consuming for tools like AutoDock Vina. In the past, there are several proposals to alleviate this problem:

- apply unsupervised clustering methods like UMAP (Hernández-Hernández & Ballester, 2023) to reduce the amount of molecules and selecting centroid molecules with high Vina score;
- separate molecules based on their chemical properties to eliminate the molecules with redundant attributes (Gorgulla et al., 2023), which fails to provide a clear quantitative understanding of the structure-activity relationship;
- employing a combination of fragments/synthons to initially dock a minimal enumeration library (Sadybekov et al., 2022). this approach may overlook potential synthons due to other repulsive regions and still necessitates the evaluation of a large number of combinations involving validated synthons and scaffolds.

More importantly, these methods cannot effectively separate the active molecules with other decoys molecules, which limits their applicability to real-world scenarios. To address these issues, we turn to Probabilistic Clustering algorithms (MPC, (Liu et al., 2022)) , which provides quantitative measurement and requires no prior knowledge to learn distance-probability mapping function automatically from multi-view distribution by consistency constraints (Liu et al., 2023).

To apply probabilistic clustering to molecules, two significant challenges need to be addressed: molecular representations and probability estimation. Drawing inspiration from (Hernández-Hernández & Ballester, 2023), we utilize Morgan fingerprints as the molecular representation and leverage the Faiss (Johnson et al., 2019) library to retrieve the k-nearest neighbors for each molecule based on hashed fingerprints. The probability between molecules is defined as the likelihood of binding to the same drug target. Additionally, previous studies have shown that successful docking of a molecule onto a protein pocket depends on certain critical fragments in the molecule fitting well with the local structure of the pocket (Sarfaraz et al., 2020). Therefore, we estimate pairwise probabilities by measuring the substructure similarity between molecules, which is further validated using statistics from Lit-PCBA/DUD-E/PubChem datasets.

Specifically, we approximate this probability with the Tanimoto similarity of molecular pairs, which is denoted as follows:

$$P(e_{ij} = 1 | \text{FP}(i), \text{FP}(j)) = \frac{\text{FP}(i) \cap \text{FP}(j)}{\text{FP}(i) \cup \text{FP}(j)}, \tag{1}$$

where $e_{ij}$ represents the event that molecule $i$ and $j$ can both dock the same protein, FP denotes the morgan fingerprint, and $\cap, \cup$ quantifies common/total substructures between two Morgan fingerprints. To perform molecular clustering based on this probability, we adopt the Fast probabilistic clustering algorithm (MPC) (Liu et al., 2022). This algorithm aims to cluster similar molecules into the same groups while maximizing the overall likelihood of all pairwise matching events, under the assumption of independence.

## 4.2 INITIAL SCREENING

Regarding the docking of a molecule dataset, the current screening algorithms often encounter the challenge of maintaining a balance between precision and computational cost. Multiple docking algorithms (Halgren et al., 2004) (Corso et al., 2022) (Eberhardt et al., 2021) and scoring functions (Singh et al., 2023) (Yang et al., 2023) (Heinzelmann & Gilson, 2021) have been employed to predict the binding affinity and pose of each small molecule within the binding site. In our study, we utilize VinaGPU+, which accelerates the popular autodock vina by 100 speedup, during the Initial

Screening stage. However, alternative or multiple docking methods can also be employed in this stage. In order to alleviate the significant computational burden associated with traversing the entire molecular library, we only dock the representative molecules that are selected from the clustering results using MPC.

In this context, we consider the initial screening from two perspectives: (i) how to ensure that the selected molecules will not deviates severely from the majority molecules in a cluster (e.g., its estimated affinity is -4.0, while the average estimated affinity in this cluster is -7.0) (ii) how to estimate a reasonable binding probability for a molecule based on its vina score.

- To address the first problem, we propose a method to reduce the risk of selecting outlier molecules by considering the overall Tanimoto similarity. Specifically, we define representative molecules as those that maximize the accumulated pairwise probability, denoted as $\sum_{j \in w} P(e_{ij} = 1)$. This criterion ensures that the representative molecule shares the highest number of common substructures with other molecules in the cluster. Empirical findings demonstrate that this approach significantly decreases the likelihood of selecting outlier samples. In practice, we adopt an iterative selection process, choosing a small number of representatives with the highest accumulated pairwise probability. To prevent redundancy, we ensure that these representatives have a similarity of no more than 0.8 with existing representatives in each cluster.

- To address the second problem, we employ isotonic regression to learn the mapping function from open labeled data, which includes proteins and molecules with active or inactive label information. We limit the maximum estimated probability to 0.3, based on extensive studies (Gorgulla et al., 2023) (Sadybekov et al., 2022), which have shown that the hit success rate with top docking scores is approximately 0.3. It is worth noting that for more reliable scoring methods, such as MMGBSA (Yang et al., 2023) or FEP (Heinzelmann & Gilson, 2021), the maximum probability can be statistically calculated using labeled samples with top scores.

It is important to note that the appropriate fusion of multi-modal docking scores obtained from different tools, each with its own theoretical basis and influencing factors, can significantly reduce the risk of cluster miss. Considering the efficient integration of various modalities of information by probabilistic clustering in multi-modal clustering tasks (Liu et al., 2022), we adopt a similar approach to amalgamate different scoring results. This process consists of two steps: (i) Initially, we convert the evaluation results obtained from each computational method into binding probabilities: $P(e_i = 1 | s_i^m)$, where $e_i$ denotes whether molecule $i$ binds to the receptor protein, and $s_i^m$ represents the estimation of the docking score for molecule $i$ generated by $m$-th method. We follow the procedure from probabilistic clustering, extracting known active molecules and decoy molecules, calculating their docking scores, and then employing monotonic regression to establish a mapping function from the docking scores to binding probabilities. (ii) Drawing inspiration from MPC, we employ the following equation to fuse probabilities generated by various algorithms, which has been proven to be an equivalent transformation assuming conditional independence among the multiple modalities:

$$P(e_i = 1 | s_i^1, s_i^2, \cdots, s_i^k) = \frac{\prod_{j \in [1, \cdots, k]} P(e_i = 1 | s_i^j)}{\prod_{j \in [1, \cdots, k]} P(e_i = 1 | s_i^j) + \prod_{j \in [1, \cdots, k]} P(e_i = 0 | s_i^j)}.$$

### 4.3 ACTIVE PROBABILISTIC SCREENING

Active Learning has been widely utilized to enhance the performance of drug screening (Yu et al., 2021). However, we observe that previous active screening methods are often limited to well-studied proteins, resulting in poor generalization ability when applied to unseen proteins (Corso et al., 2022), or requiring a considerable number of wet experiments for model retraining/fine-tuning (Warmuth et al., 2001). In contrast, we have observed that active clustering can effectively mitigate these issues. This is because active clustering does not necessitate prior knowledge about the target protein or any existing active molecules. We believe that in the refinement stage of drug discovery, incorporating active probabilistic clustering to locally adjust molecular binding probabilities can significantly enhance the recall rate of active molecules. Clustering active molecules (or molecules with high binding probabilities) can provide valuable information regarding diverse binding modes and assist in constructing pharmacophores, which are crucial for further drug optimization.

During the initial screening phase, the feedback results $L$ are sorted based on the binding probability of representative molecules. In this stage, our objective is to increase the expected recall of the top K molecules, as expressed by the following formulation:

$$E[n] = \sum_{i \in L[0:K]} P(e_i = 1). \tag{2}$$

Here, $e_i = 1/0$ indicates whether molecule $i$ successfully binds or fails to bind to the target protein. To maximize this value with a limited number of wet lab experiment results, we propose an Active Probabilistic Refinement method consisting of two key components: (i) A query strategy that selects the candidate molecule with the highest expected recall improvement. (ii) The context probability refinement mechanism.

Our query strategy quantifies the potential impact of conducting wet experiments on a molecule $i$ from cluster $W$ by evaluating the expected improvement in $E[n]$. This consideration is reasonable within the context of probabilistic molecule clustering and drug screening for two main reasons.

Firstly, our approach is built upon the fundamental assumption that active molecules are distributed within clusters. Adjusting the docking probability based on wet experiment results increases the likelihood of discovering similar active molecules. By targeting specific molecules within a cluster, we can potentially uncover additional active molecules with similar characteristics.

Secondly, effective docking tools are more likely to assign a higher affinity score to active molecules compared to inactive molecules. Consequently, clusters containing active molecules are more likely to be prioritized for investigation ahead of clusters primarily composed of inactive molecules.

To formally define the expected recall improvement, we employ the following formula:

$$\mathbb{E}[\Delta n | i] = P(e_i = 1) \sum_{j \in W} [P(e_j = 1 | e_i = 1, s_{ij}, d_j) - P(d_j = 1 | d_i, s_{ij}, d_j)], \tag{3}$$

where $W$ denotes the cluster that contains mol $i$. Eq.(3) only consider the probability gain when mol $i$ is checked to be active, since we assume there exists many candidates with similar binding probability to take the vacancy of top $i$ when mol $i$ is checked to be inactive.

As we only dock the representative molecules in each cluster at first, we are not able to calculate this formula strictly, hence we propose two variants with different granularity: molecule-based and cluster-based.

For the cluster-based, we have $\mathbb{E}[\Delta n | i] = P(e_i = 1)(1 - P(e_i = 1))|W|$. This formula approximates the difference between $P(e_j = 1 | e_i = 1, s_{ij}, d_j)$ and $P(e_j = 1 | d_i, s_{ij}, d_j)$ by $1 - P(e_i = 1)$. Intuitively, this indicates that as the probability of molecule $i$ being active increases, the probabilities of its neighboring molecules being active also tend to increase proportionally.

For the molecule-based, $P(e_j = 1 | e_i = 1, s_{ij}, d_j)$ is calculated by independent event assumption of $e_i = 0/1$, $e_j = 0/1$, $e_{ij} = 0/1$. $e_{i/j} = 1$ means $i/j$-th mol binds to the target, $e_{ij} = 1$ means $i$-th bind to the same target with $j$-th mol:

$$P(e_j = 1 | d_i, s_{ij}, d_j) = \frac{P_i S_{ij} P_j + (1 - P_i)(1 - S_{ij}) P_j}{P_j(P_i S_{ij} + (1 - P_i)(1 - S_{ij})) + (P_i(1 - S_{ij}) + (1 - P_i))(1 - P_j)} \tag{4}$$

$$\mathbb{E}_m(\Delta n | i) = P(e_i = 1) \sum_{j \in w_i} [P(e_j = 1 | e_i = 1, s_{ij}, d_j) - P(e_j = 1 | d_i, s_{ij}, d_j)]. \tag{5}$$

To select molecules for wet experiments, we follow a two-step process. Firstly, we sort the molecules based on a rapid computation of cluster-based recall gain using Equation (4.3). Next, we expand the molecules from the top clusters and dock them using Vina-GPU+. We then calculate the exact molecule-based recall gain using Equation (5). Finally, we select the molecule with the highest molecule-based recall gain for the wet lab experiment.

If the wet lab experiment confirms the activity of the selected molecule, we update its binding probability to 1. On the other hand, if the experiment shows no activity, we set the binding probability to 0. Furthermore, we update the posterior probability of the corresponding cluster using Equation (4).

It is important to note that the context probability refinement Equation (4) only covers pairwise posterior probability computation. However, it can be easily extended when there are multiple molecules with wet experiment results within a single cluster. To accomplish this, we simply enumerate all possible combinations of 0s and 1s for the neighbor molecules (limited to a maximum of 10 for computational efficiency). We then calculate the probability density and compute the posterior probability by comparing the positive events against the sum of all events' probability density.

Lastly, this process of active molecule selection, wet lab experimentation, and refinement is iteratively performed until the expected recall number meets our requirements or the number of wet lab experiments reaches the limit.

## 5 EXPERIMENTS

Our experiments are structured as follows: In Section 5.1, we present the details of our experimental setup. Next, in Section 5.2, we evaluate the effectiveness of APDD compared to a baseline method using two well-known screening datasets. Additionally, in Section 5.3, we validate our assumption that active molecules exhibit cluster-grouping behavior by analyzing statistical results from probabilistic clustering. Finally, in Section 5.4, we investigate the efficacy of APDD when the size of the molecule dataset exceeds one million.

### 5.1 EXPERIMENT SETTING

**Datasets.** We evaluate the performance of APDD using two well-known drug discovery datasets: DUD-E (Mysinger et al., 2012) containing 102 target proteins, and LIT-PCBA (Tran-Nguyen et al., 2020) consisting of 15 target proteins. To facilitate drug discovery on these proteins and the vast number of molecules, our data preparation involves three steps: (i) We extract the pocket coordinates from the existing complex structures of the target proteins. (ii) Next convert the proteins and molecules into the standard '.pdbqt' format to enable docking using VinaGPU+. (iii) Extract Morgan fingerprints of all active/decoy molecules.

After the dataset preparation, we identified 79 proteins in DUD-E and 11 proteins in LIT-PCBA that fulfilled the experiment conditions without requiring further preprocessing or encountering docking errors. Table 1 provides detailed information on these proteins.

**Baseline and Implementation.** Existing screening methods typically select molecules with the highest docking scores for wet experiments (Sadybekov et al., 2022; Gorgulla et al., 2023). To emulate this approach, we employ VinaGPU+ to dock target protein with all candidate molecules. Subsequently, we conduct wet lab experiments on molecules with the best affinity scores in a sequential manner until a predetermined recall number is achieved. We refer to this baseline scheme as Vina Enumeration (VE) for ease of reference.

Regarding the implementation of APDD, we set the number of k-nearest neighbors as 50 for Fast Probabilistic Clustering (FPC). During the Active Probabilistic Refinement procedure, we select two representative molecules from each cluster. We terminate the APDD cycle when the recall rate of the top 100 molecules reaches the target recall rate. It is worth noting that machine learning models cannot be retrained or fine-tuned due to the limited number of wet experiments. As a result, these methods are not included in the baseline comparison.

**Evaluation.** We use two different metrics to measure the performance of APDD: The reduction of Vina-GPU+ docking and wet lab experiments for discovering active molecules.

### 5.2 PERFORMANCE COMPARISON

We evaluate the performance of APDD with VinaGPU+ on these 79 target proteins of DUD-E and 11 targets of LIT-PCBA respectively. Specifically, we observe how the scale of molecule datasets, different types of proteins and the ROC of docking scoring influence the performance of APDD. We report the results on DUD-E in Table 2, and the result on LIT-PCBA in Table 3. Our observations are as follows:

- APDD achieves the same recall rate with significantly fewer VinaGPU+ docking and wet lab experiments on most of the proteins in both DUD-E and LIT-PCBA. APDD achieves 82%/75%

Table 1: The table presents the detailed experimental results for each target protein in DUD-E. The 'WLE' column indicates the number of wet lab experiments conducted, while the 'Docking' column represents the number of VinaGPU+ docking runs performed.

| target | WLE | | | Docking | | | target | WLE | | | Docking | | |
|---|---|---|---|---|---|---|---|---|---|---|---|---|---|
| | APDD | VE | per (%) | APDD | VE | per(%) | | APDD | VE | per (%) | APDD | VE | per(%) |
| aa2ar | 95 | 2430 | 3.9 | 6884 | 32032 | 21.5 | hxk4 | 32 | 269 | 11.9 | 662 | 4790 | 13.8 |
| abl1 | 132 | 1253 | 10.5 | 2006 | 10932 | 18.3 | inha | 26 | 145 | 17.9 | 322 | 2343 | 13.7 |
| ace | 139 | 1087 | 12.8 | 3540 | 17180 | 20.6 | ital | 35 | 1725 | 2.0 | 1680 | 8638 | 19.4 |
| aces | 125 | 2784 | 4.5 | 5044 | 26703 | 18.9 | jak2 | 31 | 174 | 17.8 | 910 | 6607 | 13.8 |
| adrb1 | 120 | 1087 | 11.0 | 3150 | 16105 | 19.6 | kif11 | 24 | 42 | 57.1 | 1084 | 6965 | 15.6 |
| adrb2 | 138 | 1333 | 10.4 | 2962 | 15233 | 19.4 | kit | 173 | 1173 | 14.7 | 1960 | 10616 | 18.5 |
| akt1 | 84 | 5431 | 1.5 | 3710 | 16743 | 22.2 | kith | 15 | 201 | 7.5 | 384 | 2907 | 13.2 |
| akt2 | 43 | 430 | 10.0 | 1350 | 7017 | 19.2 | kpcb | 38 | 81 | 46.9 | 1548 | 8836 | 17.5 |
| aldr | 34 | 184 | 18.5 | 1694 | 9159 | 18.5 | lck | 154 | 2363 | 6.5 | 5658 | 27820 | 20.3 |
| andr | 81 | 199 | 40.7 | 2322 | 14619 | 15.9 | lkha4 | 34 | 86 | 39.5 | 1498 | 9620 | 15.6 |
| aofb | 49 | 792 | 6.2 | 534 | 7022 | 7.6 | met | 43 | 159 | 27.0 | 2036 | 11416 | 17.8 |
| bace1 | 160 | 1340 | 11.9 | 3870 | 18383 | 21.1 | mk01 | 25 | 119 | 21.0 | 736 | 4629 | 15.9 |
| braf | 42 | 194 | 21.6 | 1816 | 10100 | 18.0 | mk10 | 51 | 1005 | 5.1 | 874 | 6712 | 13.0 |
| casp3 | 64 | 1809 | 3.5 | 1824 | 10898 | 16.7 | mk14 | 133 | 775 | 17.2 | 7782 | 36428 | 21.4 |
| cdk2 | 91 | 228 | 39.9 | 5722 | 28323 | 20.2 | nram | 32 | 860 | 3.7 | 1262 | 6298 | 20.0 |
| comt | 12 | 81 | 14.8 | 444 | 3891 | 11.4 | pa2ga | 43 | 437 | 9.8 | 954 | 5248 | 18.2 |
| cp3a4 | 242 | 1134 | 21.3 | 1802 | 11967 | 15.1 | parp1 | 70 | 243 | 28.8 | 6102 | 30558 | 20.0 |
| cxcr4 | 18 | 654 | 2.8 | 582 | 3446 | 16.9 | pgh1 | 169 | 777 | 21.8 | 1580 | 11011 | 14.3 |
| dhi1 | 88 | 785 | 11.2 | 3370 | 19679 | 17.1 | pgh2 | 65 | 92 | 70.7 | 4316 | 23614 | 18.3 |
| dpp4 | 97 | 1398 | 6.9 | 7880 | 38135 | 20.7 | plk1 | 32 | 641 | 5.0 | 1122 | 6907 | 16.2 |
| drd3 | 50 | 50 | 100.0 | 6832 | 34561 | 19.8 | pnph | 67 | 198 | 33.8 | 1608 | 7053 | 22.8 |
| dyr | 63 | 806 | 7.8 | 4332 | 17432 | 24.9 | ppara | 83 | 209 | 39.7 | 3306 | 19770 | 16.7 |
| egfr | 121 | 611 | 19.8 | 7258 | 35592 | 20.4 | pparg | 137 | 1124 | 12.2 | 4436 | 25784 | 17.2 |
| esr1 | 69 | 88 | 78.4 | 4462 | 21081 | 21.2 | ptn1 | 78 | 1872 | 4.2 | 1310 | 7380 | 17.8 |
| esr2 | 66 | 154 | 42.9 | 4548 | 20574 | 22.1 | pur2 | 20 | 90 | 22.2 | 668 | 2750 | 24.3 |
| fa7 | 50 | 342 | 14.6 | 1342 | 6364 | 21.1 | pygm | 29 | 736 | 3.9 | 602 | 4027 | 14.9 |
| fa10 | 66 | 570 | 11.6 | 4600 | 20629 | 22.3 | pyrd | 23 | 32 | 71.9 | 1224 | 6561 | 18.7 |
| fabp4 | 12 | 12 | 100.0 | 494 | 2797 | 17.7 | reni | 95 | 296 | 32.1 | 1414 | 7062 | 20.0 |
| fak1 | 51 | 192 | 26.6 | 932 | 5450 | 17.1 | rxra | 32 | 32 | 100.0 | 1422 | 7081 | 20.1 |
| fkb1a | 52 | 575 | 9.0 | 1030 | 5911 | 17.4 | sahh | 20 | 40 | 50.0 | 732 | 3513 | 20.8 |
| fpps | 120 | 1452 | 8.3 | 2162 | 8934 | 24.2 | src | 135 | 2859 | 4.7 | 7264 | 35027 | 20.7 |
| glcm | 37 | 619 | 6.0 | 538 | 3854 | 14.0 | tgfr1 | 29 | 114 | 25.4 | 1220 | 8633 | 14.1 |
| gria2 | 68 | 141 | 48.2 | 2161 | 12003 | 18.0 | thrb | 155 | 1515 | 10.2 | 6224 | 27465 | 22.7 |
| grik1 | 32 | 448 | 7.1 | 1262 | 6651 | 19.0 | try1 | 209 | 1919 | 10.9 | 5972 | 26428 | 22.6 |
| hdac2 | 144 | 319 | 45.1 | 1832 | 10486 | 17.5 | tryb1 | 42 | 593 | 7.1 | 1562 | 7798 | 20.0 |
| hivint | 52 | 567 | 9.2 | 884 | 6749 | 13.1 | tysy | 35 | 136 | 25.7 | 1116 | 6859 | 16.3 |
| hivpr | 73 | 332 | 22.0 | 3106 | 16102 | 19.3 | urok | 79 | 1437 | 5.5 | 2166 | 10012 | 21.6 |
| hivrt | 67 | 853 | 7.9 | 3064 | 19233 | 15.9 | wee1 | 25 | 25 | 100.0 | 1210 | 6251 | 19.4 |
| hmdh | 65 | 387 | 16.8 | 1642 | 8920 | 18.4 | xiap | 43 | 628 | 6.8 | 1030 | 5250 | 19.6 |
| hs90a | 27 | 595 | 4.5 | 848 | 4938 | 17.2 | – | – | – | – | – | – | – |

Table 2: The table presents the detailed experimental results for each target protein in DUD-E

| target | WLE | | | Docking | | | target | WLE | | | Docking | | |
|---|---|---|---|---|---|---|---|---|---|---|---|---|---|
| | APDD | VE | per (%) | APDD | VE | per(%) | | APDD | VE | per (%) | APDD | VE | per(%) |
| ADRB2 | 40351 | 75491 | 53.5 | 48838 | 311765 | 15.7 | ALDH1 | 499 | 730 | 68.4 | 21512 | 107237 | 20.1 |
| FEN1 | 20440 | 30101 | 67.9 | 38130 | 351078 | 10.9 | GBA | 44178 | 70468 | 62.7 | 49503 | 291404 | 17.0 |
| IDH1 | 9710 | 25543 | 38.0 | 38605 | 358796 | 10.8 | KAT2A | 36740 | 30457 | 120.6 | 37633 | 342923 | 11.0 |
| MAPK1 | 5822 | 2009 | 289.8 | 11992 | 61875 | 19.4 | MTORC1 | 1506 | 3899 | 38.6 | 2223 | 33069 | 6.7 |
| OPRK1 | 2274 | 48793 | 4.7 | 34275 | 269499 | 12.7 | PKM2 | 13465 | 10290 | 130.9 | 33640 | 245225 | 13.7 |
| VDR | 13129 | 29393 | 44.7 | 33793 | 263303 | 12.8 | – | – | – | – | – | – | – |

reduction of docking/wet experiments in DUD-E, while 85%/40% in LIT-PCBA. Moreover, we notice that APDD shows such advantage in both cases where the number of molecules is only less than 4k and where the number of molecules reaches more than 300k, which reveals that APDD is robust to the size of datasets and different chemical properties and pocket structures.

- We have observed that VinaGPU+ sometimes provides low-quality assessments for the majority of molecules, resulting in two unexpected outcomes: (i) Both APDD and VE require thousands of wet lab experiments to achieve the target recall rate. This is attributed to the extremely low AUC value of the VinaGPU+ scores, i.e., less than 0.5. (ii) APDD requires the same or even more wet lab experiments to discover the target number of active molecules. In this case, we find that VinaGPU+ scores for all molecules uniformly fall within a small range (e.g., [-7, -9] for MAPK1 and KAT2A, and [-5, -7] for PKM2), making it meaningless to compare the required amount of wet lab experiments in this scenario.

Table 3: The distribution of the clusters that contain active molecules

| | protein | $R_k$ | | | $P_k$ | | |
|---|---|---|---|---|---|---|---|
| | | $k=2$ | $k=4$ | $k=6$ | $k=2$ | $k=4$ | $k=6$ |
| DUD-E | aa2ar | 0.18 | 0.37 | 0.52 | 1.0 | 1.0 | 1.0 |
| | abl1 | 0.40 | 0.53 | 0.60 | 1.0 | 1.0 | 1.0 |
| | ace | 0.24 | 0.40 | 0.50 | 1.0 | 1.0 | 1.0 |
| | aces | 0.23 | 0.37 | 0.55 | 1.0 | 1.0 | 1.0 |
| LIT-PCBA | ADRB2 | 0.47 | 0.64 | 0.64 | 0.66 | 0.57 | 0.57 |
| | ALDH1 | 0.33 | 0.62 | 0.76 | 0.74 | 0.55 | 0.47 |
| | FEN1 | 0.39 | 0.64 | 0.76 | 0.76 | 0.56 | 0.49 |
| | GBA | 0.16 | 0.27 | 0.45 | 0.75 | 0.54 | 0.35 |

Table 4: The performance of APDD and VE on five proteins from DUD-E, where we augment the inactive molecules to over 1.4 million.

| target | WLE | | | Docking | | | target | WLE | | | Docking | | |
|---|---|---|---|---|---|---|---|---|---|---|---|---|---|
| | APDD | VE | per (%) | APDD | VE | per(%) | | APDD | VE | per (%) | APDD | VE | per(%) |
| aa2ar | 390 | 33238 | 1.2 | 398970 | 1411696 | 28.3 | adrb1 | 5922 | 54425 | 10.88 | 398892 | 1411469 | 28.26 |
| casp3 | 426 | 72965 | 0.6 | 398902 | 1411412 | 28.26 | dhi1 | 957 | 9967 | 9.6 | 398950 | 1411544 | 28.26 |
| esr1 | 39 | 176 | 22.2 | 398978 | 1411610 | 28.26 | – | – | – | – | – | – | – |

## 5.3 EXPLORATION ON MOLECULE DISTRIBUTION

In this section, we aim to explore why APDD can save the unnecessary VinaGPU+ docking and the wet lab experiments simultaneously. We report the distribution of the clusters with active molecules for four proteins in DUD-E and four proteins in LIT-PCBA as an instance in Table 3. We observe two metrics: The first is the active molecule ratio $R_k$, which is calculated between two quantities: (i) the cumulative sum of active molecules within clusters that contains no more than $k$ molecules, and (ii) the overall count of active molecules. The second is the active cluster purity $P_k$, which is calculated between two quantities: (i) the cumulative sum of active molecules within clusters that contains no more than $k$ molecules, and (ii) the total number of molecules in these clusters.

As Table 3 shows, We find that active molecules are completely separated from inactive molecules in datasets of DUD-E, and the majority of the active molecules are concentrated together in a few clusters whose size is fewer than eight. We attribute the separation between active and inactive molecule to the key assumption that the majority of active molecules are grouped together rather than uniformly distributed with the inactive molecules.

## 5.4 PERFORMANCE ON LARGE DATASETS

Previous drug discovery process is typically implemented on ligand sets with over millions of molecules. Hence, we aim to investigate the performance of APDD upon millions of molecules. Specifically, we randomly select five proteins from DUD-E and take the inactive molecules of every protein (1.4 million in total) as the inactive molecules of this selected protein. We then test APDD and VE on these augmented datasets, and report the result in Table 4. We observe that APDD still successfully recovers the preset number of active molecules with 20% VinaGPU+ docking and wet experiments compared to VE. This shows that potential of APDD in real world drug discovery applications.

## 6 CONCLUSION

In the field of early drug discovery, there is a growing trend towards using both intelligent computing and automation to drive the entire process. This new approach can be seen as an active probabilistic learning problem within the realm of machine learning. To address this, we present a novel method called active probabilistic drug discovery (APDD), which combines ligand-based screening through molecule clustering and structure-based screening using docking methods. We conducted extensive experiments on 90 targets from benchmark datasets such as DUD-E and LIT-PCBA for drug screening purposes. The results demonstrated that the APDD method achieved substantial cost savings while maintaining high screening accuracy. Our proposed paradigm aims to eliminate the need for lead optimization by significantly expanding the size of virtual molecule libraries, thereby facilitating comprehensive coverage of drug-like chemical space in well-organized clusters.

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
