# OpenReview forum: "Active Probabilistic Drug Discovery"
_ICLR.cc/2024/Conference — Submitted to ICLR 2024_

### Official Review · Reviewer_vXPn · 2023-10-22

**Soundness:** 1 poor
**Presentation:** 1 poor
**Contribution:** 1 poor
**Rating:** 1
**Confidence:** 5

**Summary:**

This paper claims to present a new method for early-stage drug discovery termed Active Probabilistic Drug Discovery (APDD). However, the paper lacks professionalism in writing, readability, and proper citation, while failing to provide sufficient information to validate the results. The proposed APDD method does not exhibit innovation algorithmically, technically, or conceptually. In sum, in its current form, I do not recommend it for publication in any rigorous academic venue. Yet, if the authors can cite properly and furnish the experiment-related details, I believe the paper could be published in a chemistry journal.

**Strengths:**

## Importance of the problem
The paper appears to address an important issue of early drug discovery.

**Weaknesses:**

## Incorrect terminology
The paper frequently refers to its method as active learning, though, from the formulation of the problem and the method's query strategy, it's set within a Bayesian Optimization framework. While the naming doesn't impact the results, it does impair readability to some extent. For consistency, I will retain the term active learning in the following text.

## Poor readability
There are several typos, such as "1030 and 1060" in the last line of page 2 and the first line of page 3. The tables in the paper do not provide sufficient information for readers to understand what the numbers represent.

## Improper citation
There is exaggerated narrative and biased citation in the introduction and related work sections. Many seminal works in the field of self-driving labs were not cited, while one paper [1] was cited multiple times, inaccurately implying that all of these concepts were first introduced in this paper [1]. When discussing the problems of active learning in drug screening, the paper mentioned poor generalization and inefficiency, but the cited references do not support these claims, leading to a misleading narrative.

## Insufficient Experimental Evidence
Despite emphasizing the integration with wet-lab experiments in the introduction and related work sections, the paper offers no explanation or evidence regarding the wet-lab experiments. It's unclear what experiments were conducted, if any.

## Incomplete Review of Related Work
Compared to traditional virtual screening (the baseline in the paper), the core change in the proposed method is the utilization of experimental results to update the model, akin to a multi-fidelity approach. The paper should at least review existing multi-fidelity methods in the related work section to provide a basis for evaluating the contribution of this work.

## Lack of Novelty
The purported APDD method, as described, is merely an application of pool-based Bayesian optimization in the drug screening scenario, with the clustering-based method being a common approach in iterative experiment design. The concept of combining wet-lab experiments for closed-loop discovery is now quite standard, and using experimental results to replace computational simulations is a straightforward multi-fidelity approach. In essence, both the technical and conceptual contributions of this paper are quite limited.

## Unremarkable Results
The well-known principle that structurally similar molecules are likely to have similar functions is not surprising to chemists. The improvement in results obtained by incorporating experimental data into simulations, compared to using simulations alone, is not astonishing. A comparison should be made with other multi-fidelity models to ascertain the significance of the results.

Reference:
[1] Wei, L., Xu, M., Liu, Z., Jiang, C., Lin, X., Hu, Y., ... & Zhang, P. (2023). Hit Identification Driven by Combining Artificial Intelligence and Computational Chemistry Methods: A PI5P4K-β Case Study. Journal of Chemical Information and Modeling, 63(16), 5341-5355.

**Questions:**

N/A

---

### Official Review · Reviewer_FDuX · 2023-10-31

**Soundness:** 2 fair
**Presentation:** 3 good
**Contribution:** 2 fair
**Rating:** 3
**Confidence:** 3

**Summary:**

This paper considers the problem of early screening for molecular discovery and proposes an algorithm for narrowing down the search space and active selection of molecules with potentially high docking probability.

The paper proposes a pipeline consisting of three steps 1) Unsupervised clustering of molecules based on the Tanimoto kernel to identify groups of molecules with similar properties 2) Choosing a representative molecule for each cluster, and discarding clusters who's representative molecule does not exhibit high docking potential 3) solving an active learning problem on the remaining clusters: iteratively and adaptively querying "promising" candidates based on the feedback received from wet experimentation.

**Strengths:**

The paper targets an important problem and as far as I understood includes experiments on real molecules (wet lab).

The proposed pipeline addresses the key issues in the early screening problem in drug discovery and proposed valid interesting solutions to them (e.g. clustering and active learning).

The paper is written well and is easy to follow.

**Weaknesses:**

### Contributions
The primary contribution of the paper is mentioned to be formulating the early drug screening problem as an active learning problem. This is widely known and used across different algorithms. Among many others, the Coley Lab (https://coley.mit.edu/research/) has done extensive research on different aspect of probabilistic (early) drug discovery through active learning and bayesian optimization among other methods. Despite the high relevance no work is mentioned.

While the overall pipeline, some choices in objective function and algorithm design are new, the modules do not bear novelty, for instance the idea of clustering molecules according to the Tanimoto kernel is previously proposed [Ramzi 2009] and the idea of active selection molecules is widely used in current literature on virtual screening and denovo design [Graff 2021, among many others]. The clustering algorithm is taken directly from Liu et al.

---
### Method
It seems to me that the method is highly relying on the choice of Morgan fingerprints and the Tanimoto kernel for clustering. Through this, the clustering is performed and regions of the domain are entirely neglected for the next steps. If these representation/clustering is not a valid choice, then the rest of the method will do poorly. I am unable to verify this since there are no comparison with other algorithms or ablation studies (e.g. clustering using a different kernel, using a different method, different choice for the representative molecule in the cluster etc.) Overall, I think the method is interesting and the idea of discarding some regions is appealing in practice, as long as we can ensure that favorable regions are not being neglected.

---
### Evaluation and Experimentation
There's no comparison with other probabilistic/active learning method for virtual screening. The experiments compare APDD with a vanilla algorithm which tests all candidate molecules. In particular I find the following experiments to be lacking (given that the ICLR standards)

- Ablation study which shows why and how every step of the pipeline is integral to achieving these results.
- Comparison with state-of-the-art active screening methods or any other baseline for that matter. In particular the MoLPAL (Molecular Pool-based Active Learning, Graff et al 2021) algorithm targets the very same problem and is a natural baseline for this method.
- Performance on common benchmarks, e.g. PMO on ZINC (Gao et al. 2022).

---
### Presentation
* The paper lacks the pseudo code of the modules appearing in their pipeline and ideally the full pipeline.
* I am not familiar with experimental work in this area, but it seems to me that more detail on the experiments must be included to make them reproducible. The paper does not have an appendix to outline the technical details of the computational/wet lab experiments.
* To publish in ICLR (as a Machine Learning venue) would be nice to include definition of some domain specific concepts, e.g. pockets, ligand, docking problem etc. Would be to good to include a glossary of chemical terms and techniques in the appendix.

---
### References

Nasr, Ramzi J., S. Joshua Swamidass, and Pierre F. Baldi. "Large scale study of multiple-molecule queries." Journal of Cheminformatics 1 (2009): 1-19.

Graff, David E., Eugene I. Shakhnovich, and Connor W. Coley. "Accelerating high-throughput virtual screening through molecular pool-based active learning." Chemical science 12.22 (2021): 7866-7881.

Gao, Wenhao, et al. "Sample efficiency matters: a benchmark for practical molecular optimization." Advances in Neural Information Processing Systems 35 (2022): 21342-21357.

**Questions:**

I have stated the ambiguities in sections "Method" and "Evaluation" of my review above.

### Typos/Small issues
* [References] The main paper is cut in the middle of the references section, some of the references e.g. [Wei et al 2023] is not included.
* [Bottom of page 2]: Size of chemical space is not 1030 - 1060, it is $10^{30}$-$10^{60}$.

---

### Official Review · Reviewer_Fua7 · 2023-10-31

**Soundness:** 2 fair
**Presentation:** 2 fair
**Contribution:** 2 fair
**Rating:** 3
**Confidence:** 2

**Summary:**

The paper proposes a model for early drug discovery called Active Probabilistic Drug Discovery (APDD), which is composed of three steps: probabilistic clustering, selective docking, and active wet experiments. The proposed method relies on the assumption that, for a given protein, the molecules that bind have features that can be clustered in the chemical space. They claim to reduce computational docking expenses by 80% and wet lab experiment costs by 70%. Their key contribution is the formulation of a pipeline for early drug discovery, where wet-lab experiments provide feedback to the computational model.

**Strengths:**

1. The paper proposes a novel early drug discovery pipeline, where the outcomes of wet lab experiments are used to update the computational model.
2. The paper claims to reduce computational docking expenses by 80% and wet experiment costs by 70%.

**Weaknesses:**

1. While the paper is relevant to ML-based drug discovery, the contributions and presented methods might be better received in a bioinformatics journal.

**Questions:**

1. Are there any particular classes of receptors that are challenging to consider in this framework?

---

### Official Review · Reviewer_LU6u · 2023-11-03

**Soundness:** 2 fair
**Presentation:** 2 fair
**Contribution:** 2 fair
**Rating:** 3
**Confidence:** 5

**Summary:**

The authors propose design of the end-to-end virtual screening method which they call  Active Probabilistic Drug Discovery (APDD). To warm up the system, the authors first cluster the molecules in a molecular database using the Probabilistic clustering algorithm which organises the data points based on the similarity of the Morgan fingerprints and selects similar molecules based on Tanimoto similarity-like scores. The representatives from the identified clusters are docked. The docking scores or other quantities such FEP scores are predicted for the remaining molecules using isotonic regression. This defines the semi-supervised learner. Next, authors introduce the query strategy designed to quantify the potential impact of conducting wet-lab experiments of candidate molecules by evaluating the expected recall improvement. The main components of the pipeline are: initial stage which warms up the learner, the clustering and sorting phase, where the molecules are organised according to their topological similarities and scored based on the predicted complex poses. Next, the molecules are scored according to the expected recall gain and the top scoring ones are selected for lab experiments. Authors postulate that their primary contribution is: ” formulating this new paradigm for early drug discovery and designing a novel machine learning algorithm that is effective and can easily be extended to accommodate an extra-large virtual molecule library”.

**Strengths:**

The problem is well chosen, well motivated and sufficiently framed into the relevant background. Authors aim to overcome sparse and noisy binding affinity data with the help of unsupervised probabilistic algorithm and incorporate query strategy based on optimising the expected recall improvement.  This mitigates the chances of selecting outliers during the query strategy and takes into account the data manifold density.

**Weaknesses:**

1) The manuscript needs substantial improvement regarding the clarity and structure:
- While general concepts are introduced, the technical ones are missing. The same ideas are repeated over various sections such as emphasising the importance of automated drug discovery systems without sufficient deep-dive into technicalities of the methods. The paper is reasonably written but there is missing clear topic separation and the ideas from different sections are repetitively spilling over.  The arguments stated in section 1-3 can be compressed in order to provide a space for the technical description.
- The description of the mathematical terms and formula is not very clear, e.g. “ The probability between molecules is defined as the likelihood of binding to the same drug target.” It is rather hard to guess what is “the probability between molecules”. It is not well explained why the formula (1) in section 4.1, p. 4, is a good choice for modelling probability. sec 4.3, initial paragraph is a summary of beliefs  and hypothetical claims without any verification, the formulas (2) and (3) are not well explained, e.g. we can guess that ‘n’ stands for recall, it is not clear what ‘d’ is.

2) The docking methods are only vaguely described and discusssed:
- Section 4.2: authors claim that multiple docking algorithms and scoring functions were used to predict binding affinity to the pocket, but there is no further analysis or at least list of the multiple algorithms and their initial setting provided. Authors argue that one of their main contributions is “cost savings 80% reduction for docking”. Given that the scores are further used in “defining a probability” in probabilistic clustering, it is important to describe these terms clearly and provide arguments on how the docking scores are transformed into the probabilities.
- In the section Experiment the authors use VinaGPU+ only. Vina is a great open-source tool which is however known for its flaws and poorer performance. The discussion and supportive experiment whcih considers the impact of the docking method and docking scoring function would be beneficial.

3. Selection of the dataset for the experiment. Experiment Designs.
- DUD-E dataset is constructed such that it is highly diverse. Bemis-Murcko clustering is used to identify similar scaffolds and pick the best performing ligands from similar scaffold clusters  while ensuring that decoys are topologically diverse to mitigate the binding opportunity. When assessing the dataset with Tanimoto similarity, the average Tanimoto is very low, supporting the claim that the molecules are highly diverse when featured by ECFPs.  This implies that the scores constructed in formula (1) are most like very near 0 majority of the time. The biases are further described and explored e.g. in https://www.ncbi.nlm.nih.gov/pmc/articles/PMC6701836/
- Authors should take the properties of the dataset into consideration and demonstrate how their method performs in comparison with other query strategies rather than benchmark only against the top highly scoring docked molecules. Even comparison against balanced RF learner which predicts affinities and selects randomly the new candidates for evaluation can lead to suprisingly great results in comparison with Vina docking scores selection.

**Questions:**

1) Can the authors reason why they decided to omit query functions utilising the model uncertainties and decided for expected recall improvement?
2) Can the authors  provide a supporting experiment demonstrating  how the proposed method compares to a standard query based on e.g. on random sampling of the molecules?
3) Can the authors provide technical analysis and supportive arguments why the choice of the probability of two molecules binding to the same target is well approximated by using Tanimoto similarity and how this satisfies the definition of probability function?
4) Can the authors provide more technical details on how the experiment in section 5.4 was constructed?  “By wet-lab” - do authors mean that they used the labels provided in the DUD-E dataset or that they conducted the experiments themselves? Did authors construct the “large dataset by using the DUD-E API to generate the decoys?

---

### Meta-Review · Area_Chair_8jHL · 2023-12-07

**Metareview:**

The authors propose a system for the early stages of drug discovery driven by ideas from chemoinformatics and active machine learning.

Although the reviewers are unified in support of this general approach to early drug discovery, they also identified a number of perceived weaknesses with the manuscript as submitted:

- a perceived lack of novelty in the approach
- some questions regarding specific choices made by the authors in designing the approach (e.g., the reliance on Morgan fingerprinting)
- a perceived lack of clarity in the discussion
- questions regarding the design of the empirical study, including a lack of relevant baselines (also missing from the discussion)

The authors chose not to address these issues during the response period.

**Justification For Why Not Higher Score:**

The reviewers identified several perceived weaknesses with the manuscript as submitted during their initial reviews, and authors did not elect to address any of the issues during the author response period.

**Justification For Why Not Lower Score:**

N/A

---

### Decision · Program_Chairs · 2024-01-16

Reject